# New Formulation towards Healthier Meat Products: *Juniperus communis* L. Essential Oil as Alternative for Sodium Nitrite in Dry Fermented Sausages

**DOI:** 10.3390/foods9081066

**Published:** 2020-08-06

**Authors:** Vladimir Tomović, Branislav Šojić, Jovo Savanović, Sunčica Kocić-Tanackov, Branimir Pavlić, Marija Jokanović, Vesna Đorđević, Nenad Parunović, Aleksandra Martinović, Dragan Vujadinović

**Affiliations:** 1Faculty of Technology Novi Sad, Bulevar cara Lazara 1, University of Novi Sad, 21000 Novi Sad, Serbia; tomovic@uns.ac.rs (V.T.); jovosavanovicdimdim@gmail.com (J.S.); suncicat@uns.ac.rs (S.K.-T.); bpavlic@uns.ac.rs (B.P.); marijaj@tf.uns.ac.rs (M.J.); 2“DIM-DIM” M.I. d.o.o, Trn-Laktaši Svetosavska bb, 78252 Trn Laktaši, Bosnia and Herzegovina; 3Institute of Meat Hygiene and Technology (INMES), Kaćanskog 13, 11040 Belgrade, Serbia; vesna.djordjevic@inmes.rs (V.Đ.); nenad.parunovic@inmes.rs (N.P.); 4Faculty for Food Technology, Food Safety and Ecology, Donja Gorica, University of Donja Gorica, 81000 Podgorica, Montenegro; aleksandra.martinovic@udg.edu.me; 5Faculty of Technology Zvornik, Karakaj 1, University of East Sarajevo, 75400 Zvornik, Bosnia and Herzegovina; dragan.vujadinovic@tfzv.ues.rs.ba

**Keywords:** *Juniperus communis* L., essential oil, sodium nitrite, dry fermented sausage

## Abstract

The effect of *Juniperus communis* L. essential oil (JEO) addition at concentrations of 0.01, 0.05 and 0.10 µL/g on pH, instrumental parameters of color, lipid oxidation (2-Thiobarbituric acid reactive substances (TBARS)), microbial growth, texture and sensory attributes of dry fermented sausages produced with different levels of fat (15 and 25%) and sodium nitrite (0, 75 and 150 mg/kg) was assessed. Reduced level of sodium nitrite (75 mg/kg) in combination with all three concentrations of JEO (0.01–0.10 µL/g) resulted in satisfying physico-chemical (color and texture) properties and improved oxidative stability (TBARS < 0.3 mg MDA/kg) of dry fermented sausages produced with 25% of fat. However, sausages produced with 0.10 µL/g of JEO had untypical flavor. No foodborne pathogens (*Escherichia coli*, *Listeria monocytogenes*, *Salmonella* spp. and sulfite-reducing clostridia) were detected in any sample throughout the storage period (225 days). The results of this study revealed significant antioxidative activity of JEO and consequently its high potential as effective partial replacement for sodium nitrite in dry fermented sausages.

## 1. Introduction

Fermented sausages have been manufactured in many countries worldwide. Currently, customers are becoming progressively aware of these meat products for their unique sensory characteristics and important health benefits [1]. Dry fermented sausages are produced using fresh or frozen meat (70–80%) and back fat (20–30%), salt, starter cultures, spices and food additives [2,3]. Owing to the relatively high level of fat and distinctive processing technology (e.g., using diverse raw materials, absence of thermal treatment), fermented sausages are highly susceptible to quality deterioration, including lipid oxidation and bacterial growth [2,3].

Lipid oxidation is one of the chief non-microbial factors in quality deterioration in meat and meat-derived products [4]. It is well known that meat products become very susceptible to oxidative deterioration due to high levels of unsaturated lipids (e.g., polyunsaturated fatty acids, phospholipids and cholesterol), a variety of oxidizing agents in the muscle tissue, the presence of metal catalysts, heme pigments, etc. Lipids (triacyl-glycerides, phospholipids and sterols) are largely spread in both the intra- and extracellular space of muscle tissue. Oxidation of lipids is a three-step radical chain reaction which involves: initiation, propagation and termination with the free radical’s formation [5]. It should be highlighted that lipid oxidation leads to loss of nutritional quality, reduced shelf life, intensified toxicity and reduction of the market value of meat and meat-derived products [4].

Spoilage (*Acinetobacter*, *Lactobacillus* spp., *Pseudomonas*, *Proteus* spp., *Enterobacter*, *Leuconostoc* spp., *Moraxella*, etc., yeasts and molds) and pathogenic (e.g., *Salmonella* spp., *Campylobacter jejuni*, *Listeria monocytogenes*, *Escherichia coli* O157:H7, *Clostridium* spp.) microorganisms can diminish the quality of meat and meat products and consequently induce numerous foodborne contaminations [6]. The growth of spoilage microorganisms causes the degradation of lipids and proteins present in meat and meat products and affects the development of unpleasant quality characteristics (e.g., discoloration, slime and gas production, off-odors and off-flavors). On the other hand, pathogenic bacteria are primarily responsible for foodborne diseases and food poisoning of meat and meat-derived products. Furthermore, in past decades, foodborne diseases have been marked as essential factors of growing public health and economic problems all around the world.

Therefore, lipid oxidation and microbiological deterioration of meat and meat products can be marked as major limitations in the modern meat industry [6].

The use of synthetic additives is one of the main approaches for preventing microbial growth and oxidative reactions in meat products [7]. Nitrites (sodium and potassium nitrite) are well known food additives and curing agents in meat processing [8]. They are officially registered as preservatives by European Union directives [9]. During the process of curing, nitrites are applied in order to improve the product’s shelf life because they efficiently suppress the growth of many harmful microorganisms and impart significant antioxidant potential to meat products [10,11]. Besides the strong preservative effect, the use of nitrites contributes to the of development of the typical reddish-pink color and flavor of cured meat products [11,12]. However, these preservatives were recently marked as unhealthy to humans because they promote the formation of carcinogenic N-nitroso-compounds [13,14].

Hence, consumers are increasingly demanding fresh, natural, and negligibly processed products with lower content of artificial additives [12,15]. Essential oils are defined as volatile oils with peculiar scents isolated from aromatic and medicinal plants by hydro-distillation or by cold pressing from citrus fruit peel. It is well known that essential oils obtained from different aromatic and medicinal plants possess a significant antioxidant and antimicrobial potential and therefore they are progressively used as natural additives in the modern food industries [16,17]. They represent the complex mixture of terpenoid compounds which can be present in different parts of herbs, particularly in their waxy channels, glands and trichomes. From a chemical point of view, essential oils are usually multipart mixtures of different organic compounds (e.g., terpenoids), aldehydes, ketones, esters, acids and alcohols, where the main constituents commonly constitute up to 85% of the essential oils, while minor compounds and trace elements constitute up to 15% [18]. Predominantly, essential oils are attracting attention as natural food additives (antioxidants and/or antimicrobials), as they are “generally recognized as safe” (GRAS) and have a wide customer acceptance [19]. Hence, several authors have investigated the application of essential oils as natural additives in dry fermented sausages [20,21,22], as well as potential replacements for nitrites in processing of cooked [12,16,23] and dry cured meat products [24].

*Juniperus communis* L. is an evergreen coniferous plant widespread throughout Europe, North America and North Asia [25]. The berries obtained from the medicinal herb *Juniperus communis* L. are conventionally well known as a strong immune system booster and powerful detoxifier [26]. *Juniperus communis* L. is most frequently used in natural remedies for respiratory infections, sore throat, arthritis, muscle aches and fatigue. It has been found that plant stems have also been used in order to prevent both short- and long-term illnesses. *Juniperus communis* L. essential oil has been assessed and established for its in vitro antiradical and antioxidant activities which are mostly dependent on its chemical shape [25,26].

Due to its strong antioxidant, antibacterial, antifungal, and anti-inflammatory properties, *Juniperus communis* L. and its essential oil are widely used in food processing, and in the pharmaceutical and cosmetic industries. Terpenoids (e.g., α-pinene, limonene and myrcene) determine the strong and distinctive aroma of juniper essential oil [27].

Recently, the application of *Juniperus communis* L. essential oil as natural additive was investigated in several studies [28,29,30]. Selim et al. [29] found that *Juniperus communis* L. essential oil added at concentrations of 0.1, 0.5, and 1% possesses a weak inhibitory effect towards *Enterococci* and *Escherichia coli* O157:H7 that were inoculated in ground beef meat, stored at a temperature of 7 °C for 14 days. However, in an earlier study, Schelz et al. [28] determined the strong antimicrobial potential of *Juniperus communis* L. essential oil against *Saccharomyces cerevisiae*. In our previous research, we found that *Juniperus communis* L. essential oil efficiently suppressed lipid oxidation and microbial growth and enhanced the color of cooked pork sausages [30].

A literature review has exposed only a few published research papers that discuss the application of essential oil as natural additive in dry fermented sausage processing. There is also a lack of data regarding the application of essential oils as sodium nitrite replacements in this type of dry cured meat product. Regarding its strong antioxidant and antimicrobial potential, we hypothesized that *Juniperus communis* L. essential oil could be used as an alternative for sodium nitrite in meat processing. Thus, the aim of this study was to assess the effect of *Juniperus communis* L. essential oil as an alternative for sodium nitrite in dry fermented sausages. For these purposes, several physicochemical (pH, color and texture), microbiological (total plate count, lactic acid bacteria) and sensory (color, odor and flavor) parameters of dry fermented sausages were determined.

## 2. Materials and Methods

### 2.1. Juniperus communis L. Essential Oil

#### GC-MS Profile of Terpenoid Compounds

*Juniperus communis* L. essential oil (JEO) was purchased from the manufacturer Herba doo (Belgrade, Serbia). JEO was kept in dark glass bottles at 4 °C prior to the experiments.

For identification of volatile terpenoids from JEO, GC-MS analysis was used according to the method described by Pavlić et al. [31]. Agilent GC890N system coupled to mass spectrometer Agilent MS 5759, with HP-5MS column (0.25 mm inner diameter and 0.25 μm film thickness, 30 m length), was applied for the characterization of terpenoid profile. Flow rate of helium was 2 mL/min. JEO was dissolved in dichloromethane (approx. 1 mg/mL) and 5 μL of solution was injected in the device with split ratio 30:1. Temperature conditions were: injector temperature 250 °C, detector temperature 300 °C, initial 60 °C with linear increase of 4 °C/min up to 150 °C. The NIST 05 and Wiley 7n data base were used for compound identification. Retention equations, which describe dependence of peak area on different concentration (*R*^2^ > 0.99), were obtained using standard compounds dissolved in dichloromethane at different concentrations (1–500 μg/mL). Results were expressed as relative percentage (%).

### 2.2. Samples

Dry fermented sausages were created with two levels (15 and 25%) of pork back fat (FC). In both obtained batters, sodium nitrite (NC) was added at three concentrations (0, 75 and 150 mg/kg). Next, each batter was divided into four parts, and into each part the corresponding concentrations of JEO (0.00, 0.01, 0.05 and 0.10 µL/g) were added. The total number of batches (B) was: FC (2) × NC (3) × JEO (4) = 24 (Figure 1). Samples were collected at different storage periods (SD) involving three randomly selected dry fermented sausages from each batch at the end of drying (0) and after 75, 150 and 225 days of storage. The total number of samples was: B (24) × SD (4) × 3 = 288.

### 2.3. Preparation of Dry Fermented Pork Sausages

Dry fermented sausages were produced in a local industrial plant (A.D. Dim-Dim, Laktaši, Bosnia and Herzegovina). Batters were produced using lean pork shoulder and pork back fat in the ratio 75:25 and 85:15%. The amounts of other ingredients were calculated in relation to raw material weight, and were as follows: NaCl (2.50%), gluconic delta-lactone (0.70%), spice mix (0.50%), dextrose (0.10%), sodium nitrite (0, 75 and 150 mg/kg) and JEO (0.00, 0.01, 0.05 and 0.10 µL/g.) The meat and back fat were minced using a cutter (Krämer & Grebe, Germany), and then the other ingredients were added and mixed with them until the required temperature (1 °C) was achieved. The sausages were stuffed in 37 mm diameter collagen casings and were placed in a climate chamber (Frigovent, Serbia) for 21 days. The processes of fermentation, smoking, drying and ripening were performed at a temperature of 14–16 °C and a relative humidity of 80–95%. Produced sausages were vacuum packed (Multivac C500, Wolfertschwenden, Germany) and stored at 15 ± 1 °C for 225 days. The proximate chemical compositions of the sausages produced with 15 and 25% of back fat at the end of drying process are presented in Appendix A.

### 2.4. Physico-Chemical Analysis

The proximate chemical composition (moisture, protein, fat and ash) was determined according to International Organization for Standardization (ISO) procedures [32,33,34,35].

The pH was evaluated using a digital pH meter Testo 205 (Testo AG, USA). Before measurement it was calibrated using standard buffers (pH = 4.00 ± 0.05 and pH = 7.00 ± 0.01 at 20 ± 2 °C). pH values were determined for three samples, from each group of dry fermented sausages, in duplicate.

Color (CIE-*LAB* values: *L**—lightness; *a**—redness; *b**—yellowness) of each sample of the dry fermented sausages was measured on fresh cross cut immediately after slicing. The *L**, *a** and *b** color coordinates were determined using a MINOLTA Chroma Meter CR-400 (Minolta Co., Ltd., Osaka, Japan) using D-65 lighting, a 2° standard observer angle and an 8-mm aperture in the measuring head [16]. Prior to measurement it was calibrated using a Minolta calibration plate (No. 11333090; Y = 92.9, x = 0.3159; y = 0.3322). Color was measured for three samples (2 cm thick) from each group of dry fermented sausages in triplicate.

The TPA (Texture profile analysis) test was conducted at room temperature using TA.XT2 Texture Analyzer (Texture Technologies Corp., Scarsdale, NY/Stable MicroSystems, Godalming, UK) equipped with a standard ⌀ 75 mm cylindrical plate. TPA parameters hardness (g), springiness, cohesiveness, and chewiness (g) were determined as described by Ikonić et al. [36]. The cylindrical shape samples (2.54 cm in diameter, 2 cm thick) were taken from the central part of the sausage, and were analyzed in two cycle compressions to 50% of their original thickness at a constant test speed of 1 mm/s. Peak force during the first compression cycle was marked as hardness. The rate at which a deformed sample goes back to its undeformed condition after the deforming force is removed was defined as springiness. The ratio of the area under the second and first curve was defined as cohesiveness. Lastly, by multiplying hardness, cohesiveness and springiness, chewiness was obtained. TPA was performed for three samples from each group of dry fermented sausages in duplicate.

Lipid oxidation of dry fermented sausages was assessed using the 2-Thiobarbituric acid reactive substances (TBARS) test according to the method of Botsoglou et al. [37], with some modifications. The final step of the extraction procedure was carried out with total volume (10 mL) of TCA (trichloroacetic acid) in ultrasonic bath XUB 12 (Grant Instruments, Cambridge, UK). Spectrophotometer Jenway 6300 (Jenway, Felsted, UK) was used for absorbance measurement at 532 nm. The results of the TBARS test were expressed as milligrams of malondialdehyde per kilogram of sample (mg MDA/kg). TBARS was determined on three samples from each group of dry fermented sausages in duplicate.

### 2.5. Microbiological Analysis

Microbiological analyses were performed on three samples from each group of dry fermented sausages in duplicate. Samples (20 g) were homogenized in 180 mL 1 g/L buffered peptone water (Merk, Darmstadt) for 10 min at 200 rpm (Unimax 1010, Heidolph, Germany) and the serial of decimal dilutions were prepared (up to 7–10). From each dilution 1 mL was placed in a sterile Petri plate and poured with appropriate media depending on the type of tested microorganisms. The following microorganisms were determined: total plate count (TPC), lactic acid bacteria (LAB), *Escherichia coli*, *Salmonella* spp., *Listeria monocytogenes* and sulfite-reducing clostridia count [38,39,40,41,42,43]. TPC was enumerated in Plate Count Agar (PCA) (Merk, Darmstadt, Germany) and incubated at 30 °C for 72 h; LAB was enumerated in de Man, Rogosa and Sharpe (MRS) Agar (Merk, Darmstadt, Germany) and incubated at 30 °C for 72 h; *Escherichia coli* was determined on Tryptone Bile Glucuronic Agar (TBX agar) (Merk, Darmstadt, Germany) after an incubation at 44 °C for 24 h; *Salmonella* spp. was determined on Xylose Lysine Deoxycholate (XLD) agar (Merk, Darmstadt, Germany) after an incubation at 37 °C for 24 h; *Listeria monocytogenes* was determined on Listeria agar acc. Ottaviani and Agosti (ALOA) (Merk, Darmstadt, Germany) after an incubation at 37 °C for 24 h; sulfite-reducing clostridia count was determined on Tryptone Sulfite Cycloserine (TSC) Agar (Merk, Darmstadt, Germany) after an incubation at 37 °C for 24–48 h under anaerobic conditions. After incubation, microscopic observation of cell morphology and biochemical tests were used for typical and atypical grown colonies identification. Results were expressed as a log number of colony forming units per gram (log CFU/g).

### 2.6. Sensory Analyses

Sensory analysis was carried out by a trained panel consisting of ten members, aged 25 to 50 years, per two sessions. All panelists work at the Faculty of Technology Novi Sad, Serbia, and have wide expertise in the sensory evaluation of foods. Panelists were trained according to methods described in ISO 8586 [44], in a sensory laboratory equipped according to ISO 8589 [45]. Evaluation of sensory attributes (color, odor and flavor) was performed using the difference-from-control test [46]. Prior to analyses, sausages were equilibrated to room temperature for about 15 min. and marked with a three-digit sample number. The sausages were sliced into 2 mm thick pieces and placed on a white porcelain plate. Consumers were firstly questioned to evaluate the control sample (without JEO and with the corresponding contents of fat and nitrite) and afterward to determine how different the coded samples were from the control one. The difference was rated on a scale from 0 to 6, where 0 = no difference; 1 = very slight difference; 2 = slight/moderate difference; 3 = moderate difference; 4 = moderate/large difference; 5 = large difference; and 6 = very large difference.

### 2.7. Statistical Analysis

The statistical program STATISTICA 13.0 (TIBCO Software Inc., Palo Alto, CA, USA) was used for data analyses. The main effects (fat content, nitrite content, JEO content and storage day) were compared. All data were expressed as mean value with their standard deviation (Stdev). The two-way, three-way and four-way interactions between these effects were also tested. Differences among treatment means were compared according to *t*-test and Duncan’s multiple range test (*p* < 0.05).

## 3. Results and Discussion

### 3.1. Chemical Profile of JEO

Chemical profile of JEO was determined by GC-MS and results are presented in Table 1.

It can be observed that β-myrcene (14.12%) was the predominant compound in JEO, obtained using the conventional technique of hydro distillation. Other compounds detected in JEO with relative percentage higher than 1% were: (1) monoterpene hydrocarbons: sabinene (9.51%), β-pinene (5.39%), α-terpinene (1.95%), *p*-cymene (3.92%), d,l-limonene (8.36%), γ-terpinene (3.38%) and α-terpinolene (2.80%); (2) oxygenated monoterpenes: 4-terpineol (6.88%); (3) sesquiterpene hydrocarbons: α-cubebene (1.22%), α-copaene (1.39%), β-elemene (3.38%), caryophyllene (3.94%), α-humulene (3.26%), germacrene D (3.81%), ledene (1.40%), α-muurolene (1.30%), α-amorphene (5.43%) and germacrene B (3.74%), while all other compounds were present in content less than 1%. Results suggested a majority of terpenoids with hydrocarbons, while a lower amount could be accounted for by the oxygenated monoterpenes and sesquiterpenes. Present results were in accordance with similar studies since it was reported that monoterpene hydrocarbons [α-pinene (31.1%), β-myrcene (16.3%), sabinene (7.5%), limonene (6.2%) and β-pinene (3.7%)] were the major compounds identified in commercial JEO [47]. Similar results were also reported by Radoukova et al. [48] and Zheljazkov et al. [49]. It should be highlighted that variations in chemical profile of JEO could be related to genetic properties, geographical origin and climate conditions. Besides conventional technique of hydro distillation, Orav et al. [50] and Marković et al. [51] investigated the possibility of using a novel technique of extraction for JEO recovery. Orav et al. [50] reported that the JEO obtained using supercritical fluid extraction (SFE) with carbon-dioxide had a lower content of monoterpenes and a higher content of sesquiterpenes compared to JEO obtained using conventional hydro distillation. On the contrary, Marković et al. [51] determined a similar chemical profile for JEOs obtained using conventional hydro distillation and novel microwave-assisted hydro distillation. Therefore, it could be assumed that supercritical fluid extraction will cause co-extraction of other lipophilic compounds which could further alter the bioactivity of these extracts. Besides that, juniper variety, geographical origin, climate and post-harvest processing could significantly affect JEO yield and chemical profile of terpenoids. Furthermore, the possibility of utilization of other juniper materials, such as the needles (leaves) during hydro distillation cannot be excluded [47].

### 3.2. pH and Instrumental Parameters of Color of Dry Fermented Sausages

The pH values of dry fermented sausages are presented in Table 2.

The fat content and storage time had a significant (*p* < 0.05) effect on the pH values. The samples produced with 15% of fat had a higher pH value. Regarding storage time, it can be observed that pH values inconsistently increased throughout storage, probably as the result of formation of amino-compounds during the proteolysis in fermented sausages [52,53]. The two-way (SD × JC), three-way (FC × NC × SD, FC × SD × JC) and four-way (FC × NC × SD × JC) interactions had a significant (*p* < 0.05–0.001) effect on the pH values (Appendix A). Values of pH ranged from 5.11 (FC = 25%, NC = 0 mg/kg, SD = 0, JC = 0.05 µL/g) to 5.63 (FC = 15%, NC = 150 mg/kg, SD = 75, JC = 0.10 µL/g). Similar results were observed by Kurćubić et al. [52] and Ozaki et al. [54] in fermented meat products.

Color is one of the key quality parameters for meat and meat products [16]. The instrumental parameters of color (*L**, *a** and *b**) are displayed in Table 2. The contents of fat, nitrite and JEO, as well as storage time, had a significant (*p* < 0.05) effect on *L** values. As expected, the samples produced with 15% of fat had lower *L** values. Moreover, storage time had the effect of decreasing *L** values, according with the findings of Pateiro et al. [55]. Finally, the addition of JEO decreased the *L** value, probably as the result of interactions among bioactive compounds of JEO (phenolics, terpenes) and myoglobin [30]. The two-way (FC × SD) and four-way (FC × NC × SD × JC) interactions were also significant (*p* < 0.05) for *L** values (Appendix A). The *L** values ranged across a wide interval from 43.92 (FC = 15%, NC = 150 mg/kg, SD = 225, JC = 0.10 µL/g) to 56.64 (FC = 25%, NC = 0 mg/kg, SD = 0, JC = 0.00 µL/g).

Fat content and storage time had a significant (*p* < 0.05) effect on the *a** values. As expected, the samples produced with 15% of fat had higher *a** values. Concerning storage time, the increasing of *a** values after the 150th day of storage can be noticed. This is in accordance with the findings of Pateiro et al. [55]. The increase of *a** values could be related to the growth of the *Staphylococcus* species [56]. Faustman and Cassens [56] reported that enzymes (NADH-cytochrome b5 reductase systems, metmyoglobin reductase and nitrate reductase) of *S. carnosus* or *S. xylosus* can alter metmyoglobin to form red myoglobin derivatives and enhance the color of meat products. Two-way (FC × NC) and three-way interactions (FC × NC × SD and FC × SD × JC) suggested a significant (*p* < 0.05–0.01) effect of using both sodium nitrite (150 mg/kg) and JEO (0.10 µL/g) for enhancing the redness of low-fat (15%) dry fermented sausages (Appendix A). The lowest (9.38) and the highest (15.95) *a** values were determined in the samples: FC = 25%, NC = 0 mg/kg, JC = 0.00 µL/g, SD = 0; FC= 15%, NC = 150 mg/kg, JC = 0.10 µL/g SD = 225. No significant (*p* > 0.05) four factor interaction was detected for the *a** value.

Fat content and storage time had a significant (*p* < 0.05) effect on the *b** values. Surprisingly, the samples produced with 15 fat had higher *b** values. After the 150th day of storage, the increase in *b** value can also be noticed. Similar findings were observed by Rubio et al. [57] for comparable meat products. Two-way (FC × SD, FC × JC, NC × JC, SD × JC), three-way (FC × NC × SD, NC × SD × JC) and four-way (FC × NC × SD × JC) interactions were significant (*p* < 0.05–0.001) for *b** values (Appendix A). The lowest (5.88) and the highest (10.17) *b** value was detected in the samples: FC = 25%, NC = 0 mg/kg, SD = 75, JC = 0.01 µL/g; FC = 15%, NC = 150 mg/kg, SD = 225, JC = 0.01 µL/g.

### 3.3. TBARS Values of Dry Fermented Sausages

Lipid oxidation is one the most important parameters of quality for meat and meat products [30]. TBARS values of dry fermented sausages are presented in Table 2. The contents of nitrite and JEO and storage time had a significant (*p* < 0.05) effect on TBARS values. The inclusion of sodium nitrite (75 and 150 mg/kg) decreased TBARS values. This was probably the result of the antioxidant activity of sodium nitrite [11]. Furthermore, Honikel [11] reported that antioxidant activity of nitrites is associated with the ability of NO to fix and stabilize heme iron (Fe) of meat myoglobin, making it unavailable to catalyze reactions of oxidation. Also, Karwowska et al. [58] reported that the reduction of nitrites, from 150 to 50 mg/kg, increased TBARS values in cooked meat products. Moreover, samples produced with the addition of JEO (0.05 and 0.10 µL/g) had lower TBARS values compared to samples produced without JEO. This is the consequence of the strong antioxidant potential of JEO. Höferl et al. [25] reported that juniper berry oil significantly prevented the formation of lipid peroxidation by-products caused by TBA. Certain compounds, such as *α*-terpinene, *γ*-terpinene and *α*-terpinolene exhibit strong antioxidant activity in prevention of lipid oxidation which could be compared with *α*-tocopherol [59]. On the other hand, certain compounds from JEO (pinene, sabinene and limonene) have a rather weak effect. Similar findings of the antioxidant effects of JEO in meat products were observed in our previous study [30]. As expected, storage time had a significant (*p* < 0.05) effect on increasing TBARS values, as the result of lipid oxidation [2]. The two-way interactions (FC × SD and NC × SD) were significant (*p* < 0.05) for TBARS values. Moreover, three-way (FC × NC × SD and FC × SD × JC) and four-way interactions had a significant (*p* < 0.05–0.001) effect on TBARS values (Appendix A). The highest TBARS value (0.398 mg MDA/kg) was observed in the sample: FC = 15%, NC = 0 mg/kg, SD = 225, JC = 0.05 µL/g. At the same time, TBARS values in the samples were: FC = 25%, NC = 75 mg/kg, SD = 225, JC = 0.01 µL/g and FC = 25%, NC = 75 mg/kg, SD = 225, JC = 0.05 µL/g amounted 0.117 and 0.110 mg MDA/kg, respectively. According to Melton [60], the TBARS value of 0.3 mg MDA/kg is marked as the threshold for rancidity of meat products. The obtained results suggested that interaction between sodium nitrite (75 mg/kg) and JEO (0.01 and 0.05 µL/g) efficiently reduced the lipid oxidation in high-fat (25%) dry fermented sausages. Regarding the strong lipo-solubility of terpenoid compounds (e.g., *β*-myrcene, sabinene, *β*-pinene, limonene) JEO possessed a higher antioxidant potential in dry fermented sausages produced with higher fat content (25%).

### 3.4. Microbiological Analysis of Dry Fermented Sausages

Total plate count (TPC) and lactic acid bacteria (LAB) of dry fermented sausages are presented in Table 3.

The contents of fat and nitrite did not exhibit a significant (*p* > 0.05) effect on TPC and LAB. In the case of JEO, the addition of this essential oil (≥0.05 µL/g) had a tendency to reduce the TPC (for 0.14 log cfu/g), but differences among the samples were not significant (*p* > 0.05). Hence, further optimization with a higher concentration is necessary. Moreover, in our previous study [30] we found that JEO addition (≥0.10 µL/g) efficiently reduced TPC in cooked pork sausages. On the contrary, Selim et al. [29] showed that JEO had no effect on the reduction of microbial growth in fresh beef meat. The antimicrobial potential of essential oil depends of its chemical shape. Generally, monoterpenes from the JEO (*α*-pinene, *β*-pinene, sabinene, *γ*-terpinene, *β*-myrcene, and limonene) are not efficient antimicrobials when applied singly [61]. However, a mixture of these compounds with the presence of other JEO constituents present as a minor content could result in additive or synergistic antimicrobial effects [62]. As mentioned, the utilization of a novel extraction technique (e.g., SFE) could be a good solution in order to improve the chemical profile of JEO [50]. Orav et al. [50] found that JEO obtained using SFE contained less monoterpenes (5.1%) and more sesquiterpenes and oxygenated sesquiterpenes (69.8%) with a higher antimicrobial potential. As expected, storage time significantly (*p* < 0.05) affected TPC and LAB. TPC decreased during the first 75 days of storage, then increased until the 150th day of storage and again decreased until the end of storage. This trend could be related to the reduction of LAB during storage, especially after the 150th day of storage. As mentioned, the population of LAB decreased throughout storage, probably as the consequences of low storage temperature (15 °C) and the exhaustion of sugar [63]. No significant (*p* > 0.05) two, three or four factor interactions were detected for both TPC and LAB (Appendix A). It can also be noticed that foodborne pathogens (*Escherichia coli*, *Listeria monocytogenes*, *Salmonella* spp. and sulfite-reducing clostridia) were not detected in any sample throughout the storage. The obtained results suggested that all treatments provided a satisfactory microbiological quality according to EU regulation [64].

### 3.5. Texture Analysis of Dry Fermented Sausages

Results of instrumental determination of texture characteristics are shown in Table 4.

As expected, fat content significantly changed (*p* < 0.05) the texture parameters (hardness, springiness, cohesiveness and chewiness). Samples with lower fat content showed a higher value of hardness and chewiness. An increase in hardness and chewiness as fat content decreases in dry fermented sausages was also reported by other authors [65,66], probably as a result of a more pronounced moisture loss in sausages with higher proportions of lean meat [66]. During the storage, hardness and chewiness values showed significant (*p* < 0.05) increase until the 150th day of storage, followed by decrease till day 225. Springiness value showed constant increase, while cohesiveness showed constant decrease until the 150th day of storage. Rubio et al. [67] reported increase of hardness, springiness, cohesiveness and chewiness of dry fermented sausage over the whole storage period (till 210 days), while Severini et al. [68] reported decrease in firmness, due to proteolysis. In the case of nitrites, it can be observed that the addition of sodium nitrite significantly (*p* < 0.05) affected the cohesiveness of dry fermented sausages. A similar finding was observed by Dong et al. [69] in cooked pork sausages. Moreover, Villaverdre et al. [70] found that the sodium nitrite addition at the levels of 75 and 150 mg/kg increased the hardness of fermented sausages. This could be related to the ability of sodium nitrite to promote protein oxidation and Schiff base formation [70]. Regarding JEO, it has been noticed that JEO addition had no impact on the texture parameters (hardness, cohesiveness and chewiness) of dry fermented sausages. Similarly, Viuda-Martos et al. [71] reported that rosemary essential oil has no effect on texture parameters of cooked sausages. Two-way (FC × SD) and four-way interactions were also significant (*p* < 0.05–0.001) for all texture parameters (Appendix A). The main texture parameters (hardness and chewiness) ranged in interval from 3539 (FC = 25%, NC = 75 mg/kg, SD = 0, JC = 0.00 µL/g) to 10,990 g (FC = 15%, NC = 0 mg/kg, SD = 150, JC = 0.05 µL/g) and from 926 (FC = 25%, NC = 0 mg/kg, SD = 225, JC = 0.05 µL/g) to 2759 g (FC = 15%, NC = 150 mg/kg, SD = 225, JC = 0.05 µL/g), respectively. Similar results for hardness of different dry fermented sausages were observed by Triki et al. [65] and Rubio et al. [67].

### 3.6. Sensory Analysis of Dry Fermented Sausages

Results of sensory analysis are shown in Table 5.

The fat content, JEO content and storage time had a significant (*p* < 0.05) effect on the sensory attribute of color. Two-way (FC × NC, FC × SD, NC × SD, FC × JC, SD × JC), three-way (FC × NC × SD, FC × NC × JC, FC × SD × JC, NC × SD × JC) and four-way interactions were also significant (*p* < 0.05–0.001) for this sensory attribute (Appendix A). Moreover, nitrite content, JEO content and storage had a significant (*p* < 0.05) effect on sensory attribute of odor. Furthermore, two-way (FC × NC, NC × JC, SD × JC) and three-way (FC × NC × JC, NC × SD × JC) interactions had a significant (*p* < 0.05–40.001) effect on odor (Appendix A). It should also be noticed that the numerical data for the sensory attributes of color and odor did not exceed the values of 2.0 (slight differences, less than 1.67 (color) and 1.78 (odor)), in any samples. Hence, the obtained results suggested that the fluctuations of fat, sodium nitrite and JEO, as well as storage time had no negative impact on these sensory attributes. Nitrite content, JEO addition and storage time had a significant (*p* < 0.05) effect on sensory attribute of flavor. Two-way (FC × JC, NC × JC, SD × JC), three-way (FC × NC × JC, FC × SD × JC, NC × SD × JC) and four-way interactions had also a significant (*p* < 0.05–0.001) effect on the flavor (Appendix A). The highest differences (>3, higher than moderate) of typical flavor were observed in the samples: FC = 25%, NC = 150 mg/kg, SD = 225, JC = 0.10 µL/g; FC = 15%, NC = 150 mg/kg, SD = 225, JC = 0.10 µL/g. Regarding JEO content of 0.05 µL/g, the highest difference (1.33) was observed in samples: FC = 15%, NC = 150 mg/kg, SD = 225, JC = 0.05 µL/g; FC = 15%, NC = 75 mg/kg, SD = 150, JC = 0.05 µL/g). This difference could be the result of interaction among the sodium nitrite and terpenoid-compounds of JEO. In our previous study we also determined that a high percentage of JEO had a significant effect on the strong aroma of cooked pork sausages [30]. Using novel extraction techniques (e.g., supercritical fluid extraction) at optimum conditions results in extracts which possess a strong antioxidant and antimicrobial potential, as well as mild flavor, which enables their application at lower concentration in meat processing [17,23].

## 4. Conclusions

Monoterpene hydrocarbon *β*-myrcene (14.12%) was the most abundant compound identified in JEO. The sausages produced with a lower fat content were significantly darker and redder (*p* < 0.05). Moreover, the values of hardness and chewiness were significantly (*p* < 0.05) higher in the samples produced with a lower fat content. The variations in the contents of nitrite and JEO had no negative impact on the color and texture parameters of dry fermented sausages. No foodborne pathogens were detected in any samples. The highest concentration of JEO (0.10 μL/g) had negative impact on flavor. The addition of JEO (0.01 and 0.05 μL/g) combined with reduced concentration of sodium nitrite (75 mg/kg) efficiently retarded the lipid oxidation of high-fat (25%) dry fermented sausages during 225 days of storage. Hence, JEO with evident antioxidant potential could be used as a partial replacement for sodium nitrite in fermented sausages processing. In order to enhance the antimicrobial potential of JEO, the usage of novel extraction technique (e.g., SFE) could be an effective solution. Further investigations are needed to analyze the synergistic effects of different natural extracts, isolated from various plant sources, on improving the quality and shelf-life of meat products.

## Figures and Tables

**Figure 1 foods-09-01066-f001:**
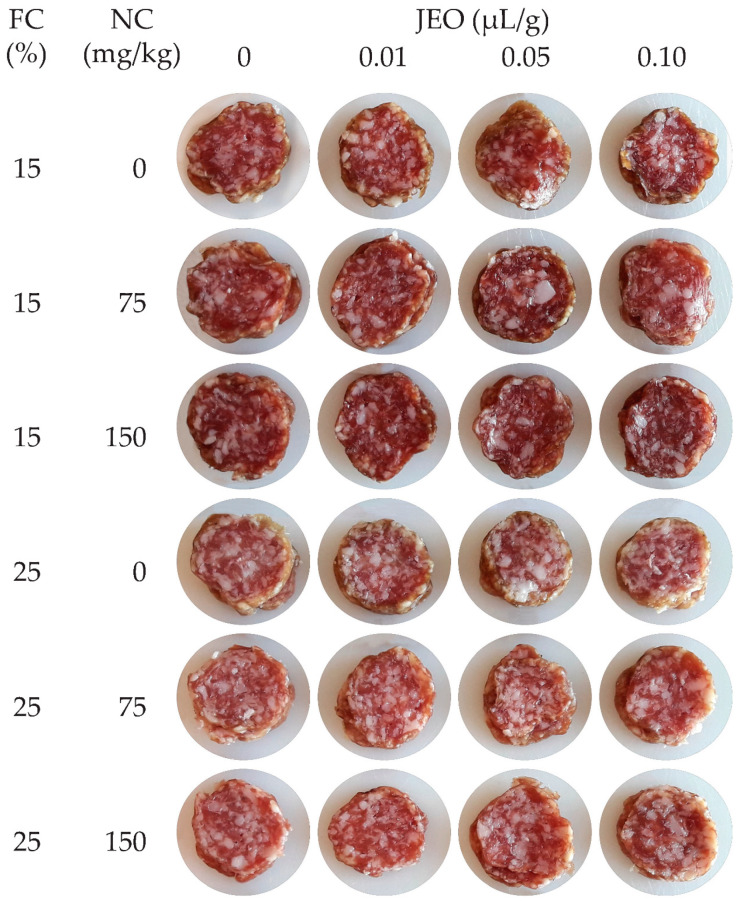
Photograph of the inside surfaces of the sausages at the end of storage.

**Table 1 foods-09-01066-t001:** Chemical profile of JEO determined by GC-MS.

	Retention Time (min)	Relative Percentage (%)
Sabinene	4.37	9.51
*β*-Pinene	4.45	5.39
*β*-Myrcene	4.71	14.12
Phellandrene	5.02	0.46
Δ-3-Carene	5.14	0.22
*α*-Terpinene	5.29	1.95
*p*-Cymene	5.49	3.92
d,l-Limonene	5.58	8.36
*γ*-Terpinene	6.33	3.38
n.i. ^1^	6.68	0.28
α-Terpinolene	7.12	2.80
Linalool	7.47	0.29
n.i.	7.66	0.12
n.i.	8.09	0.08
n.i.	8.20	0.58
*trans*-Pinocarvenol	8.60	0.47
n.i.	8.77	0.23
n.i.	8.89	0.25
Borneol	9.47	0.36
4-Terpineol	9.80	6.88
*p*-Cymen-8-ol	10.07	0.35
n.i.	10.20	1.46
Benihinal	10.31	0.24
Verbenone	10.72	0.39
n.i.	12.34	0.10
n.i.	12.71	0.27
Bornyl acetate	13.10	0.72
n.i.	13.32	0.09
n.i.	13.38	0.12
n.i.	14.33	0.22
n.i.	14.72	0.14
*α*-Cubebene	15.09	1.22
Ylangene	15.76	0.13
*α*-Copaene	15.91	1.39
n.i.	16.21	0.20
*β*-Elemene	16.45	3.38
Isoledene	16.72	0.35
Caryophyllene	17.26	3.94
Aromadendrene	17.54	0.29
*α*-Humulene	18.30	3.26
*trans*-β-Farnesene	18.44	0.86
Germacrene D	19.15	3.81
*β*-Selinene	19.29	0.17
Ledene	19.54	1.40
*α*-Muurolene	19.72	1.30
*α*-Amorphene	20.44	5.43
*γ*-Selinene	20.72	0.55
Aristolene	20.81	0.48
Germacrene B	21.36	3.74
n.i.	21.63	0.19
Spathulenol	21.99	0.62
Caryophyllene oxide	22.10	0.51
Humulene oxide	22.86	0.31
n.i.	23.01	0.38
n.i.	23.40	0.24
tau-Muurolol	23.82	0.85
*α*-Cadinol	24.18	0.99
n.i.	25.58	0.16
n.i.	25.89	0.17
Total		100

^1^ Not identified.

**Table 2 foods-09-01066-t002:** pH, instrumental parameters of color and 2-Thiobarbituric acid reactive substances (TBARS) values of dry fermented sausages.

	pH	*L**	*a**	*b**	TBARS (mg MDA/kg)
FC (%)					
15	5.46 ± 0.10 ^a^	47.8 ± 3.0 ^b^	14.0 ± 1.7 ^a^	8.01 ± 1.27 ^a^	0.17 ± 0.12 ^a^
25	5.33 ± 0.10 ^b^	52.4 ± 3.4 ^a^	12.8 ± 1.7 ^b^	7.66 ± 1.13 ^b^	0.15 ± 0.10 ^a^
*p*	<0.001	<0.001	<0.001	<0.001	0.258
NC (mg/kg)					
0	5.38 ± 0.12 ^a^	50.7 ± 4.1 ^a^	13.5 ± 1.9 ^a^	7.76 ± 1.16 ^a^	0.20 ± 0.12 ^a^
75	5.41 ± 0.12 ^a^	50.1 ± 3.7 ^a,b^	13.4 ± 1.8 ^a^	7.83 ± 1.22 ^a^	0.14 ± 0.10 ^b^
150	5.39 ± 0.12 ^a^	49.5 ± 3.9 ^b^	13.3 ± 1.7 ^a^	7.91 ± 1.27 ^a^	0.15 ± 0.10 ^b^
*p*	0.362	0.019	0.680	0.484	<0.001
JC (µL/g)					
0	5.37 ± 0.10 ^a^	51.2 ± 4.2 ^a^	13.3 ± 1.9 ^a^	7.80 ± 1.13 ^a^	0.20 ± 0.11 ^a^
0.01	5.40 ± 0.12 ^a^	49.9 ± 3.8 ^b^	13.3 ± 1.9 ^a^	7.83 ± 1.36 ^a^	0.16 ± 0.11 ^a,b^
0.05	5.40 ± 0.14 ^a^	49.8 ± 3.7 ^b^	13.4 ± 1.8 ^a^	7.82 ± 1.22 ^a^	0.14 ± 0.11 ^b^
0.10	5.41 ± 0.11 ^a^	49.4 ± 3.9 ^b^	13.5 ± 1.7 ^a^	7.89 ± 1.15 ^a^	0.14 ± 0.10 ^b^
*p*	0.316	0.001	0.759	0.924	0.003
SD					
0	5.26 ± 0.08 ^d^	50.8 ± 3.6 ^a^	13.0 ± 1.8 ^b^	7.59 ± 1.21 ^b^	0.04 ± 0.03 ^d^
75	5.47 ± 0.08 ^a^	50.3 ± 4.1 ^a,b^	13.2 ± 1.6 ^b^	7.51 ± 1.11 ^b^	0.12 ± 0.06 ^c^
150	5.40 ± 0.10 ^c^	49.7 ± 3.7 ^b^	13.2 ± 1.8 ^b^	7.72 ± 1.06 ^b^	0.20 ± 0.05 ^b^
225	5.44 ± 0.10 ^b^	49.5 ± 4.3 ^b^	14.2 ± 1.8 ^a^	8.52 ± 1.22 ^a^	0.28 ± 0.09 ^a^
*p*	<0.001	0.020	<0.001	<0.001	<0.001

FC—fat content; NC—nitrite content; JC— *Juniperus communis* L. essential oil (JEO) content; SD—storage day; Means ± Stdev with different letters ^(a–d)^ in the same column are significantly different (*p* < 0.05).

**Table 3 foods-09-01066-t003:** Microbiological quality of dry fermented sausages.

	TPC (log CFU/g)	LAB (log CFU/g)
FC (%)		
15	5.55 ± 0.73 ^a^	5.71 ± 0.87 ^a^
25	5.43 ± 0.84 ^a^	5.53 ± 0.87 ^a^
*p*	0.459	0.336
NC (mg/kg)		
0	5.51 ± 0.77 ^a^	5.64 ± 0.75 ^a^
75	5.57 ± 0.77 ^a^	5.67 ± 0.80 ^a^
150	5.40 ± 0.81 ^a^	5.55 ± 1.04 ^a^
*p*	0.705	0.851
JC (µL/g)		
0	5.51 ± 0.96 ^a^	5.62 ± 0.70 ^a^
0.01	5.65 ± 0.66 ^a^	5.68 ± 0.93 ^a^
0.05	5.44 ± 0.68 ^a^	5.55 ± 0.93 ^a^
0.10	5.37 ± 0.68 ^a^	5.63 ± 0.99 ^a^
*p*	0.642	0.968
SD		
0	5.10 ± 0.38 ^c^	6.47 ± 0.51 ^a^
75	4.74 ± 0.53 ^d^	5.81 ± 0.67 ^b^
150	6.23 ± 0.47 ^a^	5.52 ± 0.44 ^b^
225	5.91 ± 0.63 ^b^	4.68 ± 0.71 ^c^
*p*	<0.001	<0.001

TPC—total plate count; LAB—lactic acid bacteria; FC—fat content; NC—nitrite content; JC—JEO content; SD—storage day; Means ± Stdev with different letters ^(a–d)^ in the same column are significantly different (*p* < 0.05).

**Table 4 foods-09-01066-t004:** Texture parameters of dry fermented sausages.

	Hardness (g)	Springiness	Cohesiveness	Chewiness (g)
FC (%)				
15	7579 ± 1611 ^a^	0.488 ± 0.05 ^a^	0.511 ± 0.03 ^b^	1902 ± 489 ^a^
25	5282 ± 1020 ^b^	0.505 ± 0.05 ^b^	0.525 ± 0.04 ^a^	1407 ± 321 ^b^
*p*	<0.001	<0.001	<0.001	<0.001
NC (mg/kg)				
0	6271 ± 1707 ^a^	0.489 ± 0.05 ^a^	0.504 ± 0.04 ^c^	1552 ± 438 ^b^
75	6542 ± 1874 ^a^	0.500 ± 0.05 ^a^	0.519 ± 0.04 ^b^	1694 ± 508 ^a^
150	6462 ± 1712 ^a^	0.500 ± 0.05 ^a^	0.531 ± 0.03 ^a^	1713 ± 480 ^a^
*p*	0.377	0.079	<0.001	0.005
JC (µL/g)				
0	6269 ± 1752 ^a^	0.508 ± 0.05 ^a^	0.523 ± 0.04 ^a^	1654 ± 449 ^a^
0.01	6502 ± 1570 ^a^	0.492 ± 0.05 ^b^	0.516 ± 0.03 ^a^	1671 ± 467 ^a^
0.05	6672 ± 211 ^a^	0.492 ± 0.05 ^b^	0.512 ± 0.04 ^a^	1675 ± 562 ^a^
0.10	6264 ± 1556 ^a^	0.494 ± 0.04 ^b^	0.521 ± 0.04 ^a^	1612 ± 437 ^a^
*p*	0.219	0.022	0.093	0.739
SD				
0	4730 ± 915 ^c^	0.444 ± 0.04 ^c^	0.553 ± 0.04 ^a^	1153 ± 202 ^c^
75	6649 ± 1341 ^b^	0.505 ± 0.03 ^b^	0.521 ± 0.03 ^b^	1741 ± 323 ^b^
150	7518 ± 1641 ^a^	0.516 ± 0.03 ^a^	0.503 ± 0.03 ^c^	1940 ± 416 ^a^
225	6801 ± 1718 ^b^	0.521 ± 0.03 ^a^	0.496 ± 0.03 ^c^	1777 ± 506 ^b^
*p*	<0.001	<0.001	<0.001	<0.001

FC—fat content; NC—nitrite content; JC—JEO content; SD—storage day; Means ± Stdev with different letters ^(a–c)^ in the same column are significantly different (*p* < 0.05).

**Table 5 foods-09-01066-t005:** Sensory parameters of dry fermented sausages.

	Color	Odor	Flavor
FC (%)			
15	0.32 ± 0.64 ^b^	0.34 ± 0.67 ^a^	0.89 ± 1.18 ^a^
25	0.77 ± 0.77 ^a^	0.29 ± 0.68 ^a^	0.81 ± 1.12 ^a^
*p*	<0.001	0.164	0.137
NC (mg/kg)			
0	0.58 ± 0.78 ^a^	0.18 ± 0.46 ^b^	0.74 ± 1.06 ^b^
75	0.57 ± 0.77 ^a^	0.36 ± 0.73 ^a^	0.85 ± 1.17 ^a,b^
150	0.49 ± 0.67 ^a^	0.41 ± 0.77 ^a^	0.98 ± 1.20 ^a^
*p*	0.063	<0.001	0.002
JC (µL/g)			
0	0.00 ± 0.00 ^c^	0.00 ± 0.00 ^d^	0.00 ± 0.00 ^d^
0.01	0.67 ± 0.72 ^b^	0.10 ± 0.36 ^c^	0.17 ± 0.49 ^c^
0.05	0.72 ± 0.81 ^b^	0.22 ± 0.50 ^b^	0.80 ± 0.77 ^b^
0.10	0.80 ± 0.79 ^a^	0.94 ± 0.94 ^a^	2.44 ± 0.86 ^a^
*p*	<0.001	<0.001	<0.001
SD			
0	0.34 ± 0.65 ^c^	0.27 ± 0.68 ^b^	0.70 ± 1.07 ^b^
75	0.48 ± 0.67 ^b^	0.15 ± 0.41 ^c^	0.76 ± 1.09 ^b^
150	0.91 ± 0.82 ^a^	0.46 ± 0.77 ^a^	0.94 ± 1.10 ^a^
225	0.45 ± 0.69 ^b^	0.38 ± 0.73 ^a^	1.01 ± 1.29 ^a^
*p*	<0.001	<0.001	<0.001

FC—at content; NC—nitrite content; JC—JEO content; SD—storage day; Means ± Stdev with different letters ^(a–d)^ in the same column are significantly different (*p* < 0.05).

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
