# Peer review of "New Formulation towards Healthier Meat Products: Juniperus communis L. Essential Oil as Alternative for Sodium Nitrite in Dry Fermented Sausages"

_foods, 2020, doi:10.3390/foods9081066_

Round 1

Reviewer 1 Report

This manuscript from Tomovic et al. assessed the effect if Juniperus communis essential oil in dry fermented sausages. A series of parameters, such as pH and pathogenic bacteria were confirmed in this study. Overall, this study is straightforward and the results are informative. This reviewer has several comments as follows.

1) Although the JEO used in this study is commercially available, the authors should also mention and dicscuss the possibility in the manuscript that if differences among JEO batches and different extration processes from other companies. 

2) More background introduction on the application of essential oils in dry fermented sausages. What is the novelty of this study?

3)On the assessment on the color, some images should be provided.

4) On the anti-bacterial effect of JEO, the direction of the main text is not very clear and confusing and thus should be re-organized. The authors should, to some degree, explain the relevance between JEO addition and anti-bacterial effect.

Author Response

Firstly, we want to thank you for the effort you put in reviewing our manuscript and giving useful advices and comments for its improvement. We hope that we succeed in making this paper more understandable for the future readers by clarification and explanation of some important points based on your comments. All changes are marked in red color.

List of changes:

Reviewer #1:

This manuscript from Tomovic et al. assessed the effect if Juniperus communis essential oil in dry fermented sausages. A series of parameters, such as pH and pathogenic bacteria were confirmed in this study. Overall, this study is straightforward and the results are informative. This reviewer has several comments as follows.

  1. Although the JEO used in this study is commercially available, the authors should also mention and dicscuss the possibility in the manuscript that if differences among JEO batches and different extration processes from other companies.

  • According to reviewer`s suggestion we mentioned and discussed how the novel green extraction techniques affect to chemical profile of juniper essential oil (revised manuscript: lines – 251-261).

  1. More background introduction on the application of essential oils in dry fermented sausages. What is the novelty of this study?

  • We added a few sentences regarding the application of essential oil as natural additive in dry fermented sausages and as an alternative for nitrites in meat processing (revised manuscript: lines – 87-89 and 110-116). The reduction of sodium nitrite content in cured meat products is the main challenges in modern meat industry. The novelty of this study is application of juniper essential oil as novel additive and potential replacement for sodium nitrite in dry fermented sausage processing. In our further investigation we will try to develop emerging natural antioxidants, using green environmentally friendly extraction techniques (e.g. supercritical fluid extraction), which could be used as total replacement for nitrites in meat processing.

  1. On the assessment on the color, some images should be provided.

  • We added the image in supplementary material (Image S1).

  1. On the anti-bacterial effect of JEO, the direction of the main text is not very clear and confusing and thus should be re-organized. The authors should, to some degree, explain the relevance between JEO addition and anti-bacterial effect.

  • According reviewer suggestion, we revised this section in order to explain how does the JEO addition affect microbial growth in dry fermented sausages (revised manuscript: lines – 342-355).

Reviewer 2 Report

The research problem undertaken at work is interesting. The research problem undertaken is very current and well described in the introduction. The aim of the work was properly formulated.

However, in the manuscript, there are some places that must be revised. Reviewer's suggestions below:

Introduction is well written and does justify the purpose of the research.

Methodology presented in Materials and methods section does contain sufficient experimental details, however some details should be completed.

The authors assessed the quality features of sausages with different fat content as a result of a different proportion of fat raw material. It is a pity that the authors did not present in the manuscript the chemical composition of the sausages.

Lines 153 – 158 - What was the thickness of the slice of the sausage sample subjected to the color measurement? The thickness of the sample slice is decisive for the color measurement results.

Line 140 - what sugar the authors added during the production of sausages?

Results and discussion

Lines 263 – 264 - Why a* values increased after 150 days of storage?

It is surprising that the content of JEO did not inhibit the growth of bacteria, what do you think was the reason for this?

Lines 331 – 354 - the authors did not discuss the effect of nitrate content on texture parameters.

According to the reviewer, the authors should refer to other authors' research on the impact of limiting the nitrate content more often in the discussion (e.g. an article https://www.mdpi.com/2076-3921/9/1/9 or many others).

Sentence 384 - 385 in conclusions is in contrast to statements made results and discussion (lines 262) - must be reformulated

The research problem undertaken at work is interesting. The research problem undertaken is very current and well described in the introduction. The aim of the work was properly formulated.

However, in the manuscript, there are some places that must be revised. Reviewer's suggestions below:

Introduction is well written and does justify the purpose of the research.

Methodology presented in Materials and methods section does contain sufficient experimental details, however some details should be completed.

The authors assessed the quality features of sausages with different fat content as a result of a different proportion of fat raw material. It is a pity that the authors did not present in the manuscript the chemical composition of the sausages.

Lines 153 – 158 - What was the thickness of the slice of the sausage sample subjected to the color measurement? The thickness of the sample slice is decisive for the color measurement results.

Line 140 - what sugar the authors added during the production of sausages?

Results and discussion

Lines 263 – 264 - Why a* values increased after 150 days of storage?

It is surprising that the content of JEO did not inhibit the growth of bacteria, what do you think was the reason for this?

Lines 331 – 354 - the authors did not discuss the effect of nitrate content on texture parameters.

According to the reviewer, the authors should refer to other authors' research on the impact of limiting the nitrate content more often in the discussion (e.g. an article https://www.mdpi.com/2076-3921/9/1/9 or many others).

Sentence 384 - 385 in conclusions is in contrast to statements made results and discussion (lines 262) - must be reformulated

Author Response

Firstly, we want to thank you for the effort you put in reviewing our manuscript and giving useful advices and comments for its improvement. We hope that we succeed in making this paper more understandable for the future readers by clarification and explanation of some important points based on your comments. All changes are marked in red color.

Reviewer #2:

  1. The authors assessed the quality features of sausages with different fat content as a result of a different proportion of fat raw material. It is a pity that the authors did not present in the manuscript the chemical composition of the sausages.

  • According to reviewer's suggestion, we added the results of proximate chemical composition of dry fermented sausages produced with 15 and 25% of pork back fat (revised manuscript – Table S1, supplementary material).

  1. Lines 153 – 158 - What was the thickness of the slice of the sausage sample subjected to the color measurement? The thickness of the sample slice is decisive for the color measurement results.

  • The thickness of the sausage-samples for color measurement was 2 cm (revised manuscript: line - 170).

  1. Line 140 - what sugar the authors added during the production of sausages?

  • We used the dextrose (revised manuscript: lines - 148).

  1. Lines 263 – 264 - Why a* values increased after 150 days of storage?

  • The increasing of a* values could be related to growth of Staphylococcus species (Faustman, 1990) (revised manuscript: lines – 290-293).

Despite the a* values ​​increased after 150 days of storage, it should be highlighted that ΔE values ​​between all the sausages throughout the storage were lower than 2 (data not shown). Values for ΔE lower than 2 indicated the slight changes in color for meat products according to the National Bureau of Standards Unit. Generally, this study showed that the color of dry fermented sausages (with different levels of fat, sodium nitrite and JEO) were stable during a long storage period (225 days). Probably, it was the consequence of the relatively low level of lipid oxidation (TBARS <0.5 mg MDA/kg).

  1. It is surprising that the content of JEO did not inhibit the growth of bacteria, what do you think was the reason for this?

  • We revised this section in order to explain how does the JEO addition affect microbial growth in dry fermented sausages (revised manuscript: lines – 342-355). It could be assumed that monoterpenes from the JEO are not so efficient antimicrobials as it has been suggested by Dorman and Deans, 2000 (Dorman, H.J.D.; Deans, S.G. Antimicrobial agents from plants: antibacterial activity of plant volatile oils. J. Appl. Microbiol. 2000, 88, 308–316).

  1. Lines 331 – 354 - the authors did not discuss the effect of nitrate content on texture parameters.

  • According to reviewer's suggestion, we discussed the effect of sodium nitrite addition on the texture parameters of dry fermented sausages (revised manuscript: lines – 380-385).

  1. According to reviewer's suggestion, the authors should refer to other authors' research on the impact of limiting the nitrate content more often in the discussion (e.g. an article https://www.mdpi.com/2076-3921/9/1/9 or many others).

According to reviewer's suggestion we added more references in order to explain how does nitrite reduction affect quality of cured meat products (revised manuscript: lines - 87-89 and 313-315).

  1. Sentence 384 - 385 in conclusions is in contrast to statements made results and discussion (lines 262) - must be reformulated.

  • According to reviewer's, we reformulated the conclusions (revised manuscript: lines – 424-426).

Round 2

Reviewer 1 Report

The revised manuscript should be acceptable now and my only suggestion is that Image S1 can be moved to the main text since it seems important in this manuscript.

Author Response

Firstly, we want to thank you for the effort you put in reviewing our manuscript and giving useful advices and comments for its improvement. We hope that we succeed in making this paper more understandable for the future readers by clarification and explanation of some important points based on your comments. All changes are marked in red color.

List of changes:

Reviewer #1:

  1. The revised manuscript should be acceptable now and my only suggestion is that Image S1 can be moved to the main text since it seems important in this manuscript.

  • According to reviewer`s suggestion we moved the Image 1 to the main text.

Reviewer 2 Report

The authors corrected / supplemented the article as suggested by the reviewer. According to the reviewer, it is suitable for publication in its current form. I only have one comment - change the footnote to [56] - line 290.

Author Response

Firstly, we want to thank you for the effort you put in reviewing our manuscript and giving useful advices and comments for its improvement. We hope that we succeed in making this paper more understandable for the future readers by clarification and explanation of some important points based on your comments. All changes are marked in red color.

List of changes:

Reviewer #2:

  1. The authors corrected / supplemented the article as suggested by the reviewer. According to the reviewer, it is suitable for publication in its current form. I only have one comment - change the footnote to [56] - line 290.
  • According to reviewer's suggestion, we corrected this (Revised manuscript – lines – 292-293).